# Deep Prior Approach for Room Impulse Response Reconstruction

**DOI:** 10.3390/s22072710

**Published:** 2022-04-01

**Authors:** Mirco Pezzoli, Davide Perini, Alberto Bernardini, Federico Borra, Fabio Antonacci, Augusto Sarti

**Affiliations:** 1Dipartimento di Elettronica, Infomazione e Bioignegneria (DEIB), Politecnico di Milano, Via Ponzio 34/5, 20133 Milan, Italy; davide.perini@mail.polimi.it (D.P.); alberto.bernardini@polimi.it (A.B.); federico.borra@polimi.it (F.B.); fabio.antonacci@polimi.it (F.A.); augusto.sarti@polimi.it (A.S.); 2LISTENSMART S.R.L., Via Jacopo Palma 16, 20146 Milan, Italy

**Keywords:** room impulse response, convolutional neural networks (CNNs), interpolation, sound field reconstruction, inverse problems

## Abstract

In this paper, we propose a data-driven approach for the reconstruction of unknown room impulse responses (RIRs) based on the deep prior paradigm. We formulate RIR reconstruction as an inverse problem. More specifically, a convolutional neural network (CNN) is employed prior, in order to obtain a regularized solution to the RIR reconstruction problem for uniform linear arrays. This approach allows us to avoid assumptions on sound wave propagation, acoustic environment, or measuring setting made in state-of-the-art RIR reconstruction algorithms. Moreover, differently from classical deep learning solutions in the literature, the deep prior approach employs a per-element training. Therefore, the proposed method does not require training data sets, and it can be applied to RIRs independently from available data or environments. Results on simulated data demonstrate that the proposed technique is able to provide accurate results in a wide range of scenarios, including variable direction of arrival of the source, room $T_{60}$, and SNR at the sensors. The devised technique is also applied to real measurements, resulting in accurate RIR reconstruction and robustness to noise compared to state-of-the-art solutions.

## 1. Introduction

The processing of sound fields requires the acquisition of multichannel signals in order to characterize the acoustic properties of a given environment. To this end, room impulse responses (RIRs) captured with microphone arrays are particularly useful, as they model the propagation of the sound between the acoustic source and the array of microphones in the considered room. It follows that the measure of RIRs is at the basis of many tasks in sound field processing and control [1,2]. In particular, solutions to space-time acoustic field processing problems such as sound source localization [3], separation [4,5] and sound field navigation [6] take advantage of information provided by multichannel acquisitions. RIRs can also be employed for augmented and virtual reality applications, where the user explores the acoustic environment through immersive audio.

An RIR is customarily measured through specifically designed signals, e.g., sine sweeps, emitted by a loudspeaker and recorded by a microphone placed at specific points in the room [7,8,9]. In order to acquire the sound field, while satisfying the Nyquist–Shannon sampling theorem in the audible range, i.e., up to 20 kHz, the minimum distance between two sensors should be roughly lower than 1 cm. It follows that the acoustic analysis of large enclosures through the measurement of RIRs is a time-consuming and hardware demanding operation, because many sensors or a moving microphone [10] are required, also considering the fact that the measurement has to be repeated for all the source positions of interest. It follows that the aforementioned requirements constitute a limitation for practical implementations of applications where the sound field has to be known over a large region. A demand for solutions to reduce the burden of RIR acquisition and processing has therefore arisen in the research community.

Different approaches to sound field reconstruction from a reduced number of microphone measurements have been presented in the literature. The goal is either the interpolation or the extrapolation of RIRs related to locations different from those in which physical microphones or loudspeakers are placed. We can mainly identify two categories of sound field reconstruction solutions: parametric methods [11,12,13,14,15] and non-parametric [16,17,18,19,20,21] techniques. On the one hand, parametric solutions [11,12,13,14,15] rely on simplified parametric models of the sound field. Hence, the target of these techniques is to convey an effective spatial audio perception to the user rather than a perfect reconstruction of the sound field. On the other hand, non-parametric methods [16,17,18,20,21,22,23,24,25,26] aim to numerically estimate the acoustic field. In this category, the sound field is typically modelled as a linear combination of solutions of the wave equation [27], e.g., plane wave [25,26] or spherical wave [22,23,24,28,29] expansions. Alternatively, RIRs are expressed through their modal expansion [16,17] or exploiting the equivalent source method (ESM) [30]. In general, many of these techniques employ compressed sensing principles [31] in order to estimate the data from the undersampled sound field enabling the reconstruction in the target locations.

In [16] the authors adopt the common-acoustical pole and residue model originally proposed in [17] for modeling the RIR in the frequency domain, i.e., the Room Transfer Function (RTF). The model discussed in [17] splits the RTF in two components: the acoustical poles that represent the resonance frequencies of the room and the residue that depends on the source-receiver locations. From a set of microphones distributed in the room the model parameters are estimated in [16] through sequential non-linear least squares optimization. The reconstruction accuracy of this approach is limited to the low frequencies due to the inherently restricted validity of the modal representation.

Techniques based on the ESM [30] do not suffer from low frequencies limitations since the propagation is modelled using Green’s functions [27]. Approaches in this category typically rely on compressed sensing frameworks and may work directly in the time domain. In [21], the authors exploit the sparsity of the RIR signal in order to derive a dictionary of equivalent sources, defined as time-domain equivalent source method (TESM). The technique is evaluated for far-field measurements in source-free volumes.

A different approach is adopted in [18], where the sound field is decomposed into its direct and reverberant components. Under the assumption of confining all the sources in a prescribed region, the direct sound is modelled as a spatially sparse distribution of monopoles (ESM). Conversely, the reverberation is expressed as a sparse distribution of plane waves and a low-rank component. Recently, in [19] the authors expanded this approach including an explicit model of the early reflections through a sparse set of image sources.

The aforementioned techniques are built on some sparse representations in conjunction with assumptions on the properties of the acoustic space, i.e., far-field distance, absence of diffraction, and position of the sources inside the room. As long as the hypotheses are met, the methods are able to reconstruct the RIRs. However, a great variety of acoustic environments and setups exist that might differ from the assumed settings.

In [20], compressed sensing is employed to directly exploit the inherent signal structure of the RIRs without adding further assumptions. As a matter of fact, the mathematical structure of the wavefronts provides all the information related to the sound propagation in the environment. This enables in [20] the interpolation of RIRs in a Uniform Linear Array (ULA). The same strategy was previously adopted for interpolating seismic traces [32], a problem akin to the reconstruction of RIRs since it also concerns the recovery of propagating wavefronts in a medium. In practice, in [20] RIRs are modelled using a set of elementary functions, namely the shearlets [33], i.e., a dictionary of wavefronts with several sizes, orientations, and delays. The final reconstruction is thus given as the solution of a sparse optimization over the dictionary of shearlets. Although this method provides accurate results, the performance is highly affected by the background noise, as noted in [20].

Recently, deep learning [34] emerged as an effective approach for a wide range of problems in the field of acoustics [35,36,37,38,39] including RIR reconstruction. The work in [40] proposes a deep-learning-based solution to the sound field reconstruction problem, given a limited number of arbitrarily distributed measurements. The authors in [40] employ a convolutional neural network (CNN) with a UNet-like architecture [41] for recovering the RTF of a room. This solution is inspired by the literature on computer vision, where similar architectures proved to be effective for performing image inpainting [42] and super-resolution [43]. The CNN is trained over a RIR dataset of simulated rectangular rooms and tested on measurements acquired in a real rectangular room. The main limitation of this data-driven approach lies in the generalization to more complex acoustic scenarios, which would require the extension of the training set [40]. In addition, similarly to [16,17], the system in [40] is limited to the low frequencies (up to 300Hz).

In this manuscript, we propose a novel data-driven approach to the problem of RIR reconstruction. Our technique is inspired by the effectiveness of previous solutions adapted from the literature of image restoration [40] and seismic data interpolation [20]. In particular, we aim to solve the RIR reconstruction problem by adopting the so called *deep prior* approach [44,45]. Recently, Deep Image Prior (DIP) [44] has been introduced as a new framework for solving image restoration tasks (e.g., inpainting, denoising, super resolution, etc.). The same deep prior strategy has been applied also for the interpolation of seismic traces [46,47,48], vibrational data [49], and denoising of audio signals [50]. The main idea of DIP is to interpret restoration as an inverse problem, for which the adopted CNN provides a regularization prior [44]. During the optimization loop, the CNN learns a mapping from a random input to the output only by exploiting the information in the available data. Capturing the underlying structure of the data, the CNN is thus able to reconstruct the desired output in an unsupervised fashion without the need of training data sets. This flexibility comes at the cost of repeating the optimization for each target output, namely the network is trained for one output only. However, the per-element training is particularly interesting in the context of RIR reconstruction, due to the wide diversity of acoustic conditions in which one could be interested to perform RIR reconstruction e.g., reverberation time of the room, location of the sources with respect to the sensors, presence of diffraction, noise, etc.

The network architecture adopted in this work for the deep prior framework is a MultiResUNet [51]. The specific task we aim to solve is that of interpolating RIRs in a ULA, i.e., some of the channels in the array are missing. Similarly to [20], our solution works with signals in the time domain. Hence, no inherent frequency limitations are given by the system. In addition, no specific constraints on the environment or setup are imposed. However, differently from [20], where the underlying structure of the data is described using handcrafted features (shearlets), here, we exploit the representation power of CNNs to directly learn the inherent and hidden structure of RIRs from the data. Moreover, due to the regularization properties of the deep prior [45] approach, the method in this manuscript is also robust to additive noise.

The proposed method is validated through an extensive simulation campaign in Section 4. Simulated data allows us to test the performance in a wide range of scenarios synthesizing the RIRs with different characteristics. As a matter of fact, RIRs are generated varying acoustic parameters, such as the room reverberation time, the location of the sources and the signal-to-noise ratio. In addition, in order to assess the performance of the RIR reconstruction with real data, in Section 5, we adopt measured RIRs acquired using ULAs in [20]. Hence, the proposed RIR reconstruction technique based on DIP is evaluated on real measurements, comparing the results with linear interpolation and the state-of-the-art method in [20]. The results in Section 4 and Section 5 show that the proposed solution provides accurate reconstruction of RIRs independently from the various considered scenarios.

The rest of the paper is structured as follows. Section 2 introduces the signal model and the concept of RIR image; then RIR reconstruction is formulated as an inverse problem whose solution is retrieved through the DIP approach. In Section 3, the adopted architecture and the loss function are described in detail. The performance of the proposed method is assessed in Section 4 through extensive simulations. Section 5 provides results on real measurements and a comparison with the state-of-the-art technique presented in [20]. Finally, Section 6 draws conclusions and proposes some possible future works.

## 2. Problem Formulation

### 2.1. Rir Data Model

Let us consider an acoustic source located at an arbitrary position $r^{\prime }={\left[x^{\prime },y^{\prime },z^{\prime }\right]}^{T}$ and an ULA with *M* microphones. Given the distance *d* between two consecutive sensors, the position of the *i*th microphone in the ULA is defined as $r_{i}={\left[x_{a},\left(i-1\right)d,z_{a}\right]}^{T}$ with i=1,⋯,M and $x_{a}$, $z_{a}$ fixed for all the sensors in the ULA. Under the assumption that the acoustic system in which the source and the ULA are located is Linear Time-Invariant (LTI) and no noise is present, the acoustic pressure acquired by the *i*th microphone can be expressed as
(1)$$p\left(r_{i},t\right)=h\left(t,r_{i},r^{\prime }\right)*s\left(t\right),$$
where * is the linear convolution operator, *t* is the time variable, s(t) is the signal emitted by the source and $h(t,r_{i},r^{\prime })$ is the so-called RIR. The RIR describes the spatio-temporal propagation of the sound from the source at position $r^{\prime }$ to the receiver at position $r_{i}$. Due to the LTI assumption a RIR provides a complete characterization of the propagation of sound waves from the source to the sensor, therefore its knowledge is significantly important to process the acoustic field.

Different techniques for measuring the RIR between a fixed emitter and a receiver are available in the literature [8,9].

In practice, the RIRs of an ULA are sampled at sampling frequency Fs and truncated to a length of *N* samples beyond which the RIR content vanishes into the noise level. We collect the RIRs in the matrix H, here referred to as RIR image and defined as
(2)$$H=[h_{1},\cdots ,h_{M}],$$
where $h_{i}\in R^{N\times 1}$ is the vector containing the *N*-length sampled RIR of the *i*th microphone. It follows that the RIR image H in (Equation 2) has dimension N×M and, in practice, it is formed by stacking the microphone RIRs along the columns as shown in Figure 1. Given the distance *d* in the ULA, an aliasing-free sound field acquisition is bounded to a maximum frequency
(3)$$F_{\max}=\frac{c}{2d},$$
where *c* is the sound speed in air.

In order to relax the spatial-sampling condition, different strategies for the reconstruction of RIRs from an undersampled measurement set have been presented in the literature. Solutions range from compressed sensing frameworks [31] based on assumptions on the underlying signal model [16,18,21] or on the structure of the RIR image [20], to the recently proposed deep learning approaches [40].

Similarly to [20], here, we consider the problem of recovering missing RIRs from an uncomplete observation $\tilde{H}$ of the RIR image in (Equation 2) obtained as
(4)$$\tilde{H}=HS,$$
where S is a sampling operator, i.e., a M×M diagonal matrix whose elements are
(5)$$S_{i,i}=\left\{\begin{array}{cc} 1, & \mathrm{if}\;\mathrm{the}\;i\mathrm{th}\;\mathrm{channel}\;\mathrm{is}\;\mathrm{available};\\ 0, & \mathrm{otherwise}; \end{array}\right..$$

In summary, S indicates the available RIRs, since some channels of the ULA are missing in the observation $\tilde{H}$. In this manuscript, we aim at estimating the unknown data from the available channels exploiting a novel data-driven solution known as deep prior [44].

### 2.2. RIR Reconstruction Via Deep Prior

In order to interpret the RIR reconstruction task as an inverse problems, we consider the RIR image H (Equation 2) as the true model that we aim at reconstructing from the observations $\tilde{H}$ (Equation 4), i.e., the available channels of the ULA. The solution to the RIR reconstruction problem requires the inversion of the relation in (Equation 4). This is an ill-posed inverse problem, whose solution is typically constrained in order to obtain meaningful results [31]. The solution of the inverse problem ${\hat{H}}^{*}$ can be found solving the following minimization problem
(6)$${\hat{H}}^{*}=\underset{\hat{H}}{\mathrm{argmin}}\;\;J\left(\hat{H}\right)=E\left(\hat{H}S-\tilde{H}\right)+R\left(\hat{H}\right),$$
where $\hat{H}$ is the estimated RIR image, while E(·) and R(·) represent a data distance measure and a regularizer, respectively. The component E(·) provides a data-fidelity term of the estimated solution with respect to the available observation, e.g., a common choice is the mean squared error. The regularizing function R(·) expresses the a priori information on the optimal solution. In fact, due to the *ill-poseness* of the problem, multiple solutions can effectively minimize E(·) in (Equation 6) leading to match only the available data while providing poor RIR reconstruction at the missing channels. The definition of the regularizer R(·) is fundamental for finding an effective solution through the minimization in (Equation 6), and, typically, it is *handcrafted* and depending on the adopted sound propagation models, such as plane wave decomposition of the sound field [27] or ESM [30]. In the field of compressed sensing, a well-known regularization strategy is the $\ell_{1}$-norm regularization, defined in order to promote sparsity in the solution. Different techniques for RIR reconstruction adopt this approach in order exploit the inherent sparse nature of the considered sound field model [18,19,20,21,29]. It is evident that deviations from the assumed prior models in the actual data do affect the accuracy of the final estimate.

In this manuscript, we propose to adopt a deep prior [45] approach as an alternative regularization strategy for the solution of (Equation 6). Recently, in [44] the Deep Image Prior (DIP) method has been successfully employed for solving inverse problems in the context of image restoration. Similarly, DIP was applied for the reconstruction of irregularly sampled seismic [48] and vibrational [49] data, problems akin to RIR reconstruction. In this context, we consider a deep neural network as a generator described by the parametric nonlinear function f(·) so that $\hat{H}=f\left(\;\cdot \;\right)$ and the solution of the inverse problem is now given by rewriting (Equation 6) as
(7)$$\theta^{*}=\underset{\theta }{\mathrm{argmin}}\;\;J\left(\theta \right)=E\left(f\left(\theta ,Z\right)S-\tilde{H}\right),$$
where θ represents the learnable parameters of the network and Z is a random noise realization given as input. Hence, the neural network maps its parameters θ to the RIR image $\hat{H}$ given the fixed input Z [44].

By minimizing (Equation 7) rather than (Equation 6), we search for the solution to the inverse problem in the space of the neural network parameters instead of the space of the model [45]. This change of domain is at the basis of the deep prior approach. As a matter of fact, according to (Equation 7), the objective function is modified in such a way that we condition the solution through the inclusion of the prior given by the network. It is worth to underline that (Equation 7) depends solely on the data fidelity with the available observation $\tilde{H}$, hence any regularization term is replaced by the inherent prior provided by the network. Nonetheless, although the fit is performed on the observation, the implicit prior of the network reconstructs the RIRs also in the missing channels. It follows that the regularization ability of the neural network is linked to the structure of the architecture that drives the minimization (performed iteratively through gradient descent) towards solutions consistent with the prior.

In this context, convolutional neural networks (CNN) with the structure of an autoencoder [46], proved to be effective in exploiting the self-similarity in the data. In a RIR image (see Figure 1), we can observe how most of the information is shared among the channels, since the RIRs present the same inner structure, i.e., wavefronts impinging the ULA (shown as colored lines in Figure 1).

Therefore, we adopt a CNN autoencoder in order to exploit the self-similarities in the data as a prior information for the RIR image reconstruction. In fact, although the optimization (Equation 7) trains the CNN only on the available observation $\tilde{H}$, the output of the network provides an estimate of the missing ULA channels. The RIR reconstruction is achieved because the trained CNN is able to capture the inner structure in the data which is shared among the signals. The optimal reconstructed RIR image is therefore computed as
(8)$${\hat{H}}^{*}=f\left(\theta^{*},Z\right),$$
where $\theta^{*}$ are the optimal parameters that solve (Equation 7).

It is worth to underline that the deep prior paradigm needs no training data. Differently from traditional deep learning where a huge training data set is typically required to learn general features, deep prior is optimized for a single RIR image. While optimizing the network output on the available RIRs $\tilde{H}$, the structure of the CNN includes the prior information on self-similarities and correlation among the ULA channels. It follows that the choice of the network architecture represents a key aspect for the final result.

## 3. Network Description

Inspired by the effectiveness of U-Net-like networks [41] in existing deep prior applications [44,46,48,49], we decide to adopt the MultiResUNet [51] also for facing the RIR reconstruction problem discussed in this manuscript. MultiResUNet is first introduced in [51], where it shows improved results for the segmentation of multimodal medical images. U-Net-like architectures, such as MultiResUNet, are characterized by a so-called autoencoder structure as shown in Figure 2. In fact, the input of these architectures is progressively compressed by a set of convolutional layers that extract spatial features in the image in order to learn a latent representation of the data. This first process is known as encoding and the corresponding part of the neural network takes the name of encoder (block E in Figure 2). Subsequently, in the decoder D, the learned features are expanded back in order to reconstruct the target network output. Typically, the decoder is symmetric with respect to the encoder and each layer of the encoding section is connected with the corresponding decoding layer through skip connections [41].

The main feature of MultiResUNet is the possibility to work with different shapes and scales. This feature proved to be effective for the reconstruction of seismic traces in [46]. This is desirable also for the problem of RIR reconstruction, since it would allow us to exploit the self-similarity of the data at different time scales and among the channels. In MultiResUNet, the standard convolutional layers are replaced by multiresolution blocks (MultiRes). Such blocks are inspired by the inception architecture [52], which analyses the input features at different scales varying the kernel size. The block scheme of MultiRes block is depicted in Figure 2. MultiRes layers are composed of a sequence of small convolutional 3×3 blocks. The sequence of small convolutional layers effectively approximate greater 5×5 and 7×7 convolutional blocks [51] with the advantage of limiting the number of learnable parameters. Another important feature of MultiResUnet concerns the skip connections. Skip connections are a key element of UNet architectures that enables the feature sharing between corresponding layers in encoder and decoder. In MultiResUNet, skip connections are replaced by Residual Path blocks (see Figure 2). As explained in [51], discrepancy between features in the encoder and decoder blocks generates the so-called *semantic gap*. In fact, the features at the encoder that are passed unprocessed to the decoder are supposed to present lower level information since they come from earlier stages of the processing. In the decoder, conversely, the features are derived from the higher level representation of the bottleneck and possible previous decoding layers. Hence, the idea of Residual Path blocks is to process the features from the encoding layers to the corresponding decoding layers through two additional convolutional layers in order to reduce their *semantic gap* improving the network performance.

In our work the implementation of MultiResUNet adopts 3 MultiRes layers in the encoder with the corresponding decoding layers connected through Residual paths. The downsampling is performed through 3×3 convolutional layers with 2×2 stride. On the other side, the upsampling is achieved with bilinear interpolation. Throughout the network, we adopted batch normalization and LeakyReLU [53] as nonlinear activation function after each convolutional layer.

In order to find the optimal weights $\theta^{*}$ of the network for the RIR reconstruction (Equation 8), we train the MultiResNet by minimizing the $\ell_{1}$ distance between the estimate and the observed RIR image through the Mean Absolute Error (MAE)
(9)$$L=\frac{1}{MN}\sum_{m=1}^{M}\sum_{n=1}^{N}\left|{\left[\left(\hat{H}-\tilde{H}\right)M\right]}_{mn}\right|=\frac{1}{MN}\sum_{m=1}^{M}\sum_{n=1}^{N}\left|{\left[\left(f\left(\theta ,Z\right)-\tilde{H}\right)M\right]}_{mn}\right|,$$
where M is a M×M selection matrix through which we select the available data. The matrix M allows us to evaluate the loss function only on the observed channels of the RIR image since the points on the diagonal associated to missing signals are equal to zero. As a matter of fact, the deep prior approach evaluates the distance measure (Equation 7) only on the available data. At the same time, the network itself acts as a regularizer since it also provides an estimate of the missing channels achieving the reconstruction of the RIR image. As far as the input noise Z is concerned, we adopt a N×M×128 tensor of white Guassian noise with zero mean and standard deviation 0.1. The network is implemented in PyTorch [54] and trained using the Adam optimizer [55] for 3000 epochs with learning rate set to 0.01. Similarly to [44,48,49], in order to reinforce convergence at each iteration the input tensor is perturbed with additional zero-mean white noise of variance 0.03.

In Figure 3, we can observe the procedure of $\hat{H}$ generation. The output of the network is shown at different iterations of the training process. From Figure 3 it is possible to note how the network progressively reconstructs the RIRs. Starting from a rough estimation (blurred wide lines), the resolution of $\hat{H}$ progressively increases with the iterations, until the image is sufficiently detailed to discern wavefronts.

## 4. Numerical Analysis

In this section, we validate the proposed deep prior (DP) approach to RIR reconstruction analyzing the interpolation performance on simulated data. This allow us to compare the reconstruction provided by DP to the synthetic and noiseless ground truth of the simulations. Moreover, using simulations we can flexibly change the setup in terms of source positions and acoustic conditions of the environment. We consider three different scenarios with the aim of evaluating the performance of the DP by varying the Direction Of Arrival (DOA) of the sources, the reverberation time ($T_{60}$) of the room, and the Signal-to-Noise Ratio (SNR) of the signals. In particular, we simulated commercially available ULAs [56] constituted by M=32 microphones with sensors interspace d=3 cm. The maximum frequency satisfying (Equation 3) is $F_{\mathrm{MAX}}=5.716\;\mathrm{kHz}$. The two considered source configurations are shown in Figure 4. The microphone array and the sources are located in a room of dimensions 5.5 m×3.4 m×3.3 m with $T_{60}=0.6\;s$ unless otherwise stated. With the setup in Figure 4a, we simulate the situation where the ULA receives the source signal from different DOAs. Hence, we would like to observe if there exists a significant dependency of the RIR reconstruction with respect to the DOA of the source. In the second setup (see Figure 4b), the sources are located on the broadside of the array in order to simulate the typical setting of applications like teleconferencing, soundbar, etc. The sources in Figure 4b are placed at different distances in order to cover a wide area of the room. Therefore, this setup represents a general scenario in which we also varied the $T_{60}$ of the room and the SNR at the microphones.

The RIRs between microphones of the ULA and sources have been generated using the image source method [57,58] with 8 kHz sampling rate and $c=343\;ms^{-1}$. The stacks of the simulated RIRs constitute the RIR images H (Equation 2). The observation $\tilde{H}$ is obtained from H by selecting 16 channels, with an undersampling factor of 50%. Similarly to [20], the sampling operator S is random, with some constraints imposed a posteriori. In particular, the indexes of the missing channels are chosen following a uniform distribution checking that S has not to remove more than 5 consecutive microphone channels.

Figure 5 shows an example of target RIR image H, observation $\tilde{H}$, and reconstruction $\hat{H}$ obtained through the proposed DP approach. Note that only the first 9 ms of the RIRs are shown in the first row of Figure 5 in order to highlight the reconstruction performance around the direct path and the first reflections. In the second row of Figure 5, the later parts of the RIRs (ranging from 70 ms to 100 ms), characterized by higher echo density, are depicted. It is possible to observe that the DP is able to reconstruct the missing channels in $\hat{H}$ (Figure 5c). In particular, the network fills the gaps in $\tilde{H}$ consistently with respect to the surrounding context. As a matter of fact, comparing H and $\hat{H}$ in Figure 5, we can appreciate how the wavefronts (curved patterns in the RIR images) are effectively reconstructed in the missing channels. In summary, DP performs an interpolation, thus generating an estimate of the missing data, by exploiting only the information in the available channels of $\tilde{H}$ and the prior provided by the network as explained in Section 2.2.

In order to asses the performance of the RIR reconstruction, we evaluate the Normalized Mean Square Error (NMSE) between the estimates and the actual RIRs defined as [20]
(10)$$\mathrm{NMSE}\left(\hat{H},H\right)=10\log_{10}\frac{1}{M}\sum_{i=1}^{M}\frac{{∥{\hat{h}}_{i}-h_{i}∥}^{2}}{{∥h_{i}∥}^{2}}$$
where ${\hat{h}}_{i}\in R^{N\times 1}$ is the estimate of the *i*th RIR contained in the *i*th column of $\hat{H}$.

In addition to the evaluation of the reconstruction error with respect to the ground truth data, we consider as a baseline comparison the commonly adopted Linear Interpolation (LI) technique. Similarly to DP, LI does not impose assumptions on the signal model and on the acoustic setting, but in practice it implements a low pass filtering in the wavenumber-frequency domain as explained in [20].

In Figure 4a, the setup used for testing different DOAs (θ) of the source is depicted. In particular, we simulate 10 sources located on a semicircle in front of the ULA. The sources are positioned in such a way that DOAs go from $\theta =10{}^{\circ }$ to $\theta =170{}^{\circ }$ with respect to the center of the array (see Figure 4a).

Figure 6a shows the NMSE (Equation 10) of RIR reconstruction as a function of the DOA (θ). Inspecting Figure 6a, we can observe that the DP reconstruction consistently achieves lower NMSE with respect to LI. It is possible to note that the best reconstructions are achieved for both LI and DP with sources in front of the ULA. However, although the NMSE of the proposed technique has higher variance than LI, the difference among the results is limited within 3.2 dB with maximum NMSE=−5.6 dB at $\theta =10{}^{\circ }$ and a minimum of NMSE=−8.8 dB at $\theta =110{}^{\circ }$. Therefore, from the results in Figure 6a we cannot observe a significant dependency of the reconstruction performance on the source DOA.

In order to perform an analysis on the robustness to $T_{60}$ of the room and SNR at the sensors, we consider the setup in Figure 4b. In this case, J=12 sources are arranged on a grid broadside the ULA. The distance between the sources and the ULA ranges from 0.3 m to 2.1 m. This allows us to cover a wide area of the room in a region of interest for different microphone array applications, e.g., source separation, speech enhancement and localization.

The RIRs are simulated varying the $T_{60}$ of the room from 0.2 s to 0.8 s. This range of reverberation time reflects the $T_{60}$ values typically observed in office-like rooms. The adopted values are in line with the $T_{60}$ of the measured RIRs in [20] and in Section 5. The results in Figure 6b are computed averaging the NMSE obtained reconstructing the RIRs for each of the sources in Figure 4b. The average NMSE is shown as a function of the $T_{60}$ in Figure 6b, where it is possible to observe that the reconstruction provided by DP outperforms the LI estimation up to more than 4 dB. Interestingly, the reconstruction performance of DP constantly increases with the $T_{60}$. The higher reverberation energy, corresponding to longer $T_{60}$, positively affects the estimates of DP. As a matter of fact, in order to reconstruct the missing channels, DP exploits the texture in the RIR image given by the higher echo density. This property can be noted also in Figure 3, where the dense late reflections are fitted before the sparse early reflections and the direct path.

Finally, we evaluate the robustness of the proposed DP approach to noise. In particular, we simulate an additive white Gaussian noise corrupting the RIRs. The variance of the noise component is set in such a way that a desired SNR is obtained. We varied the SNR from 0 dB to 40 dB. Similarly to the analysis on $T_{60}$, we average the NMSE over all the sources in Figure 4b.

The average NMSE is reported in Figure 6c as a function of the adopted SNR values. Consistently with previous results, also in this context, the LI is outperformed by DP excluding only the case with SNR=0 dB. In fact, when SNR=0 dB the energy of the additive noise signal equals that of the RIRs, and the two techniques performs comparably with a difference of ≈0.88 dB only. Interestingly, DP can achieve low NMSE values up to 5 dB of SNR showing high robustness to additive white Gaussian noise.

## 5. Experimental Analysis

In this section we assess the performance of the proposed method with real data. This gives us the opportunity to compare it with a state-of-the-art technique in RIR reconstruction, namely the compressed-sensing-based solution (CS) presented in [20]. In particular, we apply the proposed DP method on real measurements provided in [20]. The authors in [20], acquired the RIRs in three different rooms adopting a ULA of M=100 sensors with distance d=3 cm between two consecutive microphones. The rooms considered in [20] are named as “Balder”, “Freja” and “Munin”, and the estimated $T_{30}$ are 0.32 s, 0.46 s and 0.63 s, respectively. Differently from simulations, the measured RIRs were acquired in rooms of non-convex geometries and they inherently present scattering and diffraction phenomena. The interested reader is referred to Section 5 of [20] for a detailed description of the measurement equipment and setups.

The observations $\tilde{H}$ (Equation 4) are obtained using two undersampling factors. In fact, in [20], the RIRs are randomly selected in order to have 20 or 33 channels available, corresponding to 20% and ≈30% of the ULA signals, respectively. Moreover, the undersampling operation is performed to have maximum gaps in $\tilde{H}$ of 24 cm in the 20 channels observations and 15 cm when 33 channels are available. Additive white Gaussian noise is then added to the RIRs varying the SNR at the sensors from 50 dB to 20 dB.

In Figure 7, the NMSE of the reconstructions is reported for the three rooms and undersampling factors. As expected, the performance of both CS and DP decreases when a lower number of sensors are available in $\tilde{H}$. In particular, for DP we can observe a constant offset of ≈3 dB between the NMSE of the two cases.

Inspecting Figure 7, we can note that DP produces consistent estimates regardless the SNR, differently from CS, which exhibits a decrease in the considered SNR range. The behavior of DP is in line with the analysis shown in Section 4. For instance, in room Balder (Figure 7a), the NMSE of DP is on average ≈−7.3 dB and ≈−3.7 dB for 33 and 20 available microphones, respectively. Similar results can be observed for room Freja (Figure 7b), while the performance for room Munin (Figure 7c) is around 1 dB better. Although the estimates of the two techniques are comparable at SNR=50 dB, the results of CS rapidly deteriorate with lower SNR values. At SNR=20 dB, the NMSE of CS is already ≈−0.5 dB for all the rooms. In addition, the decrease of the reconstruction performance is independent from the number of available channels.

## 6. Conclusions

In this manuscript we proposed a novel method for the reconstruction of room impulse responses based on deep neural networks. We adopted a MultiResUNet, an architecture that employs multiple resolution levels in the convolutional layers in order to exploit the similarities between the multichannel RIRs for the reconstruction. The RIR interpolation of a uniform linear array is framed as an inverse problem for which the proposed CNN provides a regularized solution through the deep prior approach [44]. As a matter of fact, while training the network to reconstruct the available data, an estimate of the interpolated RIRs is obtained thanks to the inherent prior given by the network itself. Differently from classical deep learning solutions, this procedure does not rely on standard training, but it requires a per-element training, avoiding the need of huge data sets.

We evaluate the RIR reconstruction performance of the proposed method on both simulated and measured RIRs. The performance of the deep prior method is compared to linear interpolation of the RIRs and a state-of-the-art solution based on compressed sensing [20]. The results on synthetic data show that the proposed method is able to provide accurate reconstructions under different scenarios concerning the direction of arrival, the $T_{60}$ of the room and the SNR of the signals. Results on measured RIRs shows that the proposed approach can be applied on real data. In addition, the CNN exhibits robustness to noisy measurements with more consistent results with respect to the state-of-the-art compressed sensing method.

Future works will address the extension of the devised method to arrays of different geometries. Moreover, we foresee the application of the deep prior approach to similar sound field processing problems, including for instance, sound field separation and navigation.

## Figures and Tables

**Figure 1 sensors-22-02710-f001:**
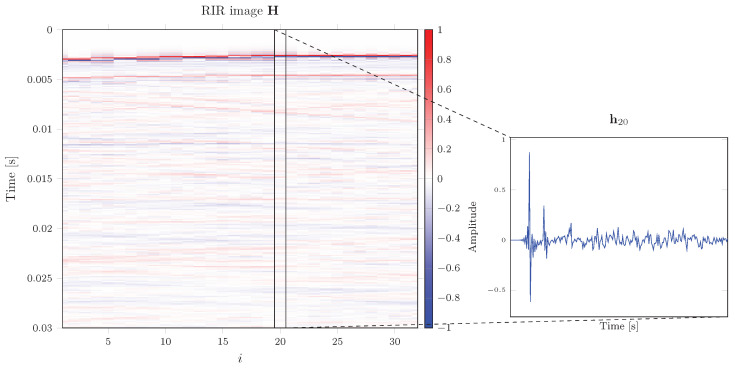
Example of a RIR image. The RIRs of the microphones are arranged along the columns of H.

**Figure 2 sensors-22-02710-f002:**
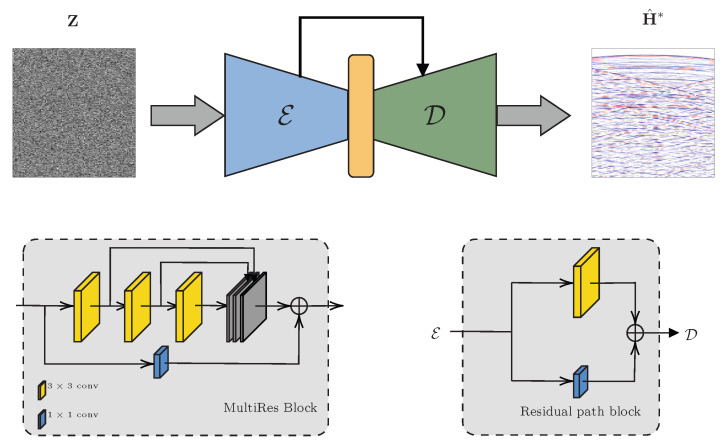
The overall block scheme of the Unet-like structure of the network, along with the block diagrams of MultiRes and Residual path blocks.

**Figure 3 sensors-22-02710-f003:**
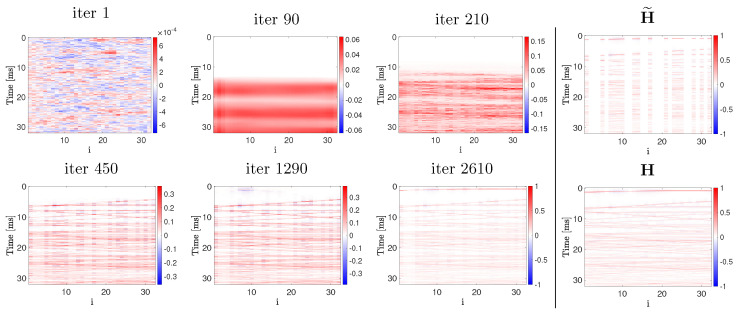
RIRs reconstruction procedure. The network output $\hat{H}$ is shown at various iterations. The observation $\tilde{H}$ and the ground truth RIR image H are reported for reference.

**Figure 4 sensors-22-02710-f004:**
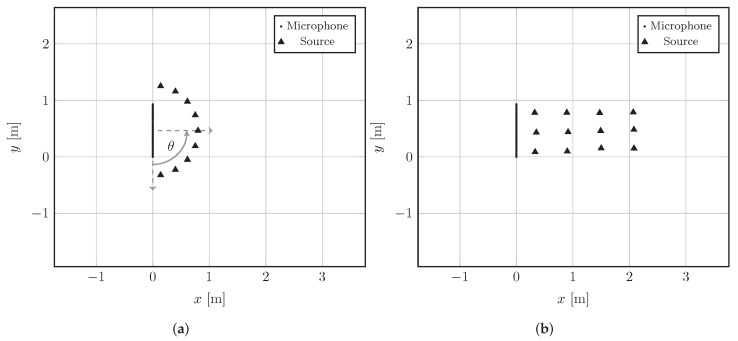
2D graphical representation of the simulated setups (top view). The microphones of the ULA are depicted as black dots, while the sources are represented as black triangles. The elements are scaled with respect to the room dimensions, hence the distance between adjacent microphones of the ULA is barely noticeable. (**a**) Variable DOA (θ) and (**b**) 12 sources setups.

**Figure 5 sensors-22-02710-f005:**
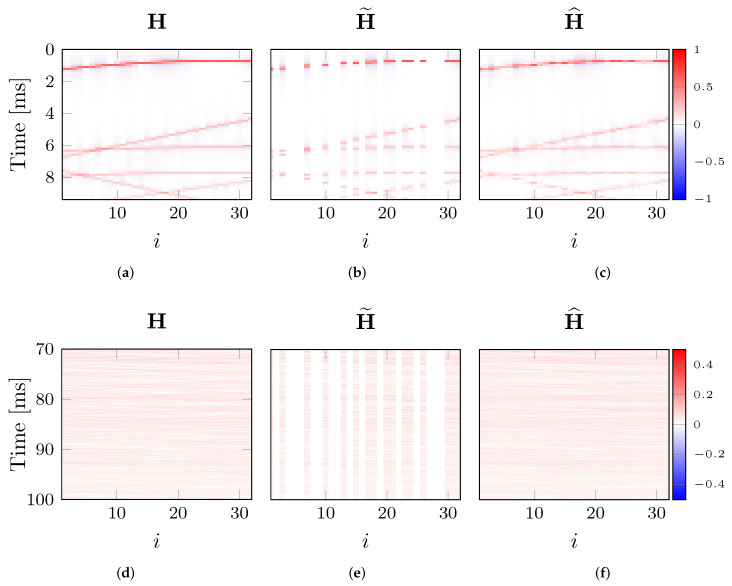
Example of a RIR image H (**a**,**d**), the observed data $\tilde{H}$ (**b**,**e**) and the reconstruction $\hat{H}$ obtained through the proposed technique (**c**,**f**). (**a**–**c**) The time axis is zoomed around the direct path and the first reflections. (**d**–**f**) The time axis is zoomed between 70 ms and 100 ms.

**Figure 6 sensors-22-02710-f006:**
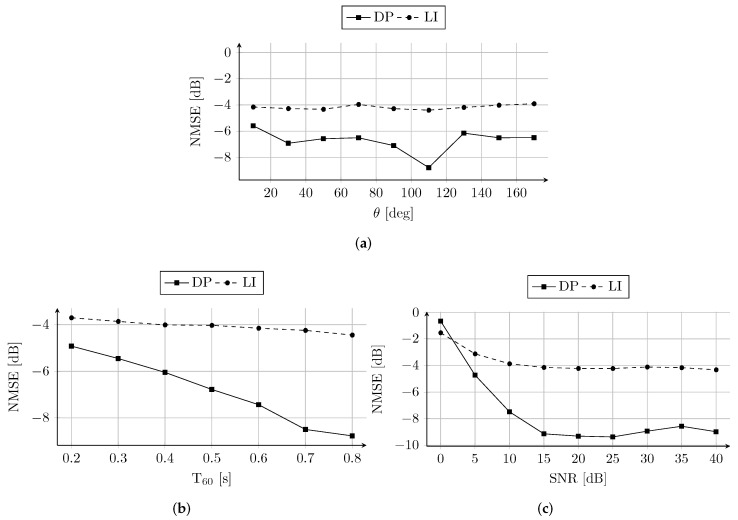
(**a**) NMSE as a function of the DOA (θ) of the source. (**b**) Average NMSE as a function of the $T_{60}$ of the room and (**c**) SNR of the sensors.

**Figure 7 sensors-22-02710-f007:**
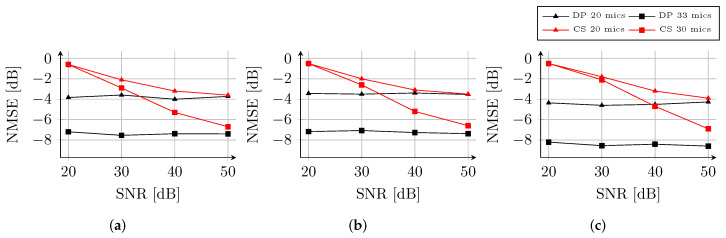
NMSE of the RIR reconstruction as a function of the SNR for rooms Balder (**a**), Freja (**b**) and Munin (**c**) from [20]. The RIR interpolations are obtained using 20 (triangle marks) or 33 (square marks) microphones. Adapted from Ref. [20], with permission from Elsevier (2022).

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
