# Peer review of "Deep Prior Approach for Room Impulse Response Reconstruction"

_sensors, 2022, doi:10.3390/s22072710_

Round 1

Reviewer 1 Report

The reconstructed room impulse response should be compared with measured one...

The precision and accuracy should be added in addition to describe the  model...

Which is the most important parameter for RIR reconstruction?

Parameters  in time and frequency domain determined from modelled, reconstructed and measured room impulse responses should be also described?

Precision and accuracy of method should be explained better...

Reviewer 2 Report

This paper investigates RIR reconstruction by using CNN. The investigated approaches are interesting, and the validation provides the efficiency and accuracy of the proposed method. Please check the following comments indicating minor revision.

Figure 4
It is hard to distinguish between each dot indicating microphone positions. Please modify the figure.

Sect.4
line 337
The sampling rate is indicated as 8kHz however, F_max is indicated as 5.716 kHz. If the sampling rate is 8kHz, I think the F_max should be under 4kHz. Please clarify.

Figure 5
The reviewer couldn't understand why each part of the wavefront lines of (b) are not "dots" but "lines"? The simulation is performed by setting discrete receiving points. Please clarify.

Figure 5
Please add the figures like the present Fig. 5 but a figure with the time duration for the late reflection in the impulse response. This figure is only showing the initial parts, and I think it is much easier to interpolate but the late parts are more difficult to interpolate due to many reflections. So, it is profitable to know how the latter reflections are interpolated.

Eq. 10
Please add what the capital letter M indicates.

Figure 6
Please clarify why the accuracy of estimation decreases as the rev time becomes shorter. Indeed, it is easy to think that the accuracy increases as the rev time becomes shorter. 

Figure 6(c)
The reviewer couldn't understand the interpolation accuracy is varied due to SNR. Please clarify the reason for this.

Reverberation time limit
In the validation part of this paper, only the conditions with the rev time up to 0.8 s are treated. How is the upper limit of the reverberation time that can be analyzed by the proposed method? Please clarify or investigate the limitation.

Round 2

Reviewer 1 Report

The authors have explained all open questions.